# The Interrelations between Biological and Targeted Synthetic Agents Used in Inflammatory Joint Diseases, and Obesity or Body Composition

**DOI:** 10.3390/metabo10030107

**Published:** 2020-03-13

**Authors:** Eric Toussirot

**Affiliations:** 1INSERM CIC-1431, CHU de Besançon, Centre d’Investigation Clinique Biothérapie, Pôle Recherche, 25000 Besançon, France; etoussirot@chu-besancon.fr; Tel.: +33-3-81-21-89-97; 2Fédération Hospitalo-Universitaire INCREASE, CHU de Besançon, 25000 Besançon, France; 3CHU de Besançon, Rhumatologie, Pôle PACTE (Pathologies Aiguës Chroniques Transplantation Éducation), 25000 Besançon, France; 4Département Universitaire de Thérapeutique, Université de Bourgogne Franche-Comté, UFR des Sciences Médicales et Pharmaceutiques de Besançon, CS 71806, 25030 Besançon Cedex, France; 5INSERM UMR1098, Relations Hôte Greffon Tumeurs, ingénierie cellulaire et génique, Université de Bourgogne Franche-Comté, 25000 Besançon, France

**Keywords:** obesity, body composition, DEXA, biological agent, JAK inhibitor, treatment response

## Abstract

Obesity is a comorbidity that plays a role in the development and severity of inflammatory joint diseases, including rheumatoid arthritis, psoriatic arthritis and axial spondyloarthritis. The relationships between obesity and adipose tissue and the treatments given for inflammatory joint diseases are bidirectional. In fact, biological agents (bDMARDs) and targeted synthetic agents (tsDMARDs) may influence body weight and body composition of treated patients, while obesity in turn may influence clinical response to these agents. Obesity is a prevalent comorbidity mainly affecting patients with rheumatoid arthritis (RA) and psoriatic arthritis (PsA) with specific phenotypes. Tumour necrosis factor alpha (TNFα) inhibitors have been associated with changes in body composition by improving lean mass, but also by significantly increasing fat mass, which localized toward the abdominal/visceral region. The IL-6 inhibitor tocilizumab is associated with an increase in lean mass without change in fat mass. The clinical response to TNFα inhibitors is attenuated by obesity, an effect that is less pronounced with IL-6 inhibitors and the B-cell depletion agent rituximab. Conversely, body weight has no influence on the response to the costimulation inhibitor abatacept. These effects may be of help to the physician in personalized medicine, and may guide the therapeutic choice in obese/overweight patients.

## 1. Introduction

Inflammatory joint diseases (IJD) are complex disorders characterized by chronic inflammation of the joints and related skeletal structures. They comprise a wide range of disorders of which rheumatoid arthritis (RA) and spondyloarthritis (SpA) are the most common. 

Rheumatoid arthritis (RA) is a systemic immune-mediated disease with chronic symmetrical joint inflammation, leading to progressive clinical deformations, erosive radiographic changes and disability. Pro-inflammatory cytokines such as IL-1β, TNFα and IL-6, but also IL-17A play a major role in the joint inflammation of RA [1]. Spondyloarthritis encompasses a group of inflammatory disorders mainly affecting the axial skeleton, i.e., the spine and the sacroiliac joints, but also specific anatomical structures such as the entheses. Peripheral arthritis may also be observed in SpA. According to the main skeletal involvement, SpA is classified as axial or peripheral disease. The most common forms of SpA are ankylosing spondylitis (AS), the prototypical form of axial SpA (ax-SpA), and psoriatic arthritis (PsA). Shared clinical manifestations in SpA are axial joint inflammation, enthesitis, dactylitis and peripheral oligoarthritis. Axial disease in SpA may lead to ligamentous ossifications, reduced spinal mobility and thus progressive ankylosis and disability [2]. TNFα and IL-17A have been identified as key mediators driving inflammation in ax-SpA, while IL-23 and IL-17A, together with TNFα are involved in PsA [3]. 

Extra-articular features may also be observed in RA and SpA. In RA, systemic inflammation can affect different organs including the eyes, skin, lungs and kidneys. However, it is well established that the cardiovascular (CV) system is predominantly involved in RA with the development of atherosclerotic CV disease [4]. This increased CV risk may be explained by the combination of traditional CV risk factors, but also by the influence of systemic inflammation. In SpA, extra-articular involvement involves the eyes (acute anterior uveitis), the skin (psoriasis) and the bowels (inflammatory bowel diseases [IBD]). However, the CV system is also often affected, and like RA, ax-SpA and PsA are also characterized by a high mortality rate linked to CV comorbidities [5]. Systemic inflammation also plays a predominant role in this enhanced CV risk. 

Inflammatory joint diseases are further characterized by obesity and changes in body composition. The interest garnered by obesity in IJD can be explained by the influence of obesity on the development of arthritis, but also by the contribution to systemic inflammation that adipose tissue may provide [6]. In turn, adipose tissue may impact on the therapeutic response to specific targeted drugs that are commonly used in the management of these diseases, i.e., biological disease-modifying antirheumatic drugs (bDMARDs) and targeted synthetic DMARDs (tsDMARDs). 

In this review, we aim to discuss the interrelations between obesity, fat mass/adipose tissue and the specific treatments that are used in clinical practice in RA and SpA. We will also discuss the influence that biological and targeted synthetic drugs may have on body composition and conversely, the role that body mass index (BMI) may play in therapeutic response to these agents. The influence of weight or BMI on the response to conventional synthetic (cs) DMARDs is not examined in this review. 

## 2. Obesity, Body Composition and Inflammatory Joint Diseases

Obesity and overweight are defined by the World Health Organization (WHO) as abnormal or excessive fat mass that presents a risk to health. Body mass index is the most common anthropometric measurement to assess and define obesity. According to this definition, BMI categorizes subjects into 3 groups: normal weight (BMI < 25 kg/m^2^), overweight (BMI between 25 and 30 kg/m^2^) and obesity (BMI > 30 kg/m^2^). However, BMI gives an evaluation of total weight, but does not distinguish between fat and lean mass, and thus, is an approximation of fat percentage. In addition, there are different phenotypes of obesity, the most commonly described being android and gynoid obesity. Waist circumference is a valid measurement for abdominal, or android, obesity, a fat distribution strongly associated with CV complications [7]. Body mass index is also influenced by several parameters, including age, gender, ethnicity and inflammation, as observed in IJD [8]. Fat mass, and other tissues, namely skeletal muscle, solid organs, skin and bone, collectively termed fat-free mass or lean mass, may be evaluated by specific techniques, given body composition measurements. The two methods that are widely used for the measurement of body composition compartments are bioelectric impedance and dual X-ray absorptiometry (DEXA) [8]. DEXA is more accurate and thus considered to be the reference technique for body composition measurements. In addition, DEXA provides measurements of total lean and fat mass, as well as the precise anatomical localisations, such as the android and gynoid regions [9].

Body composition is altered in IJD. In RA, a condition with systemic inflammation and release of pro-inflammatory cytokines with catabolic properties, a loss of body cell mass and lean mass was initially reported under the term “rheumatoid cachexia” [10]. Body cell mass includes muscle mass and visceral mass as well as immune cells. Loss of body cell mass in RA has been estimated at between 13% and 15%, therefore influencing muscle strength, energy metabolism and susceptibility to infections. TNFα has been identified as a key mediator for body mass depletion [10]. Initial reports focused on depletion of lean body mass and cachexia but more recent research has drawn attention to excess fat mass or obesity in RA [6,8]. According to the WHO definition, obesity was reported to be prevalent in 18–31% of patients and overweight in 60% of patients [6]. In fact, loss of body cell mass is usually accompanied by increased fat mass and thus stable or slightly increased body weight, leading to a phenotype of cachectic obesity in RA. Abnormal body composition, characterized by lean body mass deficit and fat mass excess has been described in patients with RA. Body composition measurements by DEXA in patients with RA showed that fat distribution was altered, with a clear shift to the abdominal region and a parallel decrease in lean mass [11]. The parallel loss of skeletal muscle and increase in fat defines body composition phenotypes such as sarcopenia, overfat and sarcopenic obesity. The different studies evaluating body composition in RA found that such phenotypes were very common in patients with long disease duration [12]. They showed abnormal values for all measures of adiposity, including abdominal fat. In a case-control study, we reported that women with RA had increased visceral (intra-abdominal) adipose tissue [13], a fat accumulation that is strongly associated with adverse CV events [7]. 

Obesity in SpA has been less widely investigated. A recent study performed in the Netherlands found that obesity was prevalent in 22% of patients, and overweight in 37% [14]. Conversely, the link between obesity and psoriasis, with or without arthritis, is well established: obesity is a factor that plays a role in the development of psoriasis and contributes to the severity of this skin disease [15]. Several studies have consistently underlined a significant link between BMI and psoriasis, while additional studies provide some evidence that increased adiposity is a risk factor for incident psoriasis [16]. An increased prevalence of obesity has also been reported in PsA as compared to the general population. The prevalence of obesity was found to be higher in patients with PsA compared to patients with psoriasis [17], and even higher than that observed in RA and psoriasis (respectively 37%, 27% and 29%) [18]. There is additional evidence linking obesity to the risk of incident PsA [19]. Collectively, obesity is associated with both psoriasis and PsA and is considered to be a contributing factor to the development of both diseases. A limited number of studies have evaluated body composition in patients with psoriasis using DEXA. In a series of Brazilian patients with psoriasis, total body fat was similar between psoriatic patients and healthy controls. In that study, patients with PsA had more evidence of adiposity with a significantly higher body fat percentage than psoriatic patients and healthy controls. In these patients, the excess of fat tissue was located in the android region [20]. Using bio-impedance, post-menopausal women with PsA were characterized by sarcopenia and increased total fat mass [21]. 

## 3. Influence of bDMARDs and tsDMARDS on Body Composition in Inflammatory Joint Diseases

We are now in the era of targeted drugs for the treatment of many immune-mediated diseases including IJD. These agents may be classified as biological agents (bDMARDs) or targeted synthetic agents (tsDMARDs). bDMARDs control the biological properties of specific pro-inflammatory cytokines (TNFα, IL-6, IL-17A, IL-23 or IL-1) or the function of a specific cellular subset (B lymphocyte or T lymphocyte co-stimulation). Available agents are TNFα inhibitors (TNFi: infliximab, etanercept, adalimumab, certolizumab and golimumab), a B-cell depletion agent (rituximab), a co-stimulation blocking agent (abatacept), IL-6 inhibitors (IL-6i: tocilizumab and sarilumab), IL-17A inhibitors (IL-17Ai: secukinumab and ixekizumab) and an IL-23p19 inhibitor (IL-23p19i: ustekinumab) [22]. Beside bDMARDs, tsDMARDs were recently introduced on the market, and especially, kinase inhibitors that have been licensed in the treatment of different immune-mediated diseases. Targeted synthetic DMARDs differ from csDMARDs by the selectivity of their action, targeting specific signaling pathways. The Janus kinase family is a small family of receptor-associated tyrosine kinases that are essential for the signal cascade of type I and type II cytokine receptors, and thus, Janus kinase inhibitors (JAKi) block the cytokine-mediated signaling pathway [23]. JAKi controls the production of several cytokines involved in immune reaction and they have proven to be highly effective in the treatment of RA and PsA. Currently, two JAKi are available, namely tofacitinib and baricitinib [24]. In RA and PsA, both bDMARDs (TNFi, IL-6i, rituximab and abatacept for RA; TNFi, IL-17Ai, IL-23p19i for PsA) and tsDMARDs (JAKi: tofacitinib and baricitinib for RA; tofacitinib for PsA) are very effective in the control of joint inflammation and progression of the disease. In ax-SpA, only bDMARDs (TNFi and IL-17Ai) are approved, leading to effective control of joint symptoms and certain extra-articular manifestations. These agents control the inflammatory process and the biological functions of specific cytokines with catabolic properties (for instance, TNFα), and may therefore potentially influence weight, BMI and/or body composition.

### 3.1. bDMARDs

#### 3.1.1. TNF Inhibitors

Initial reports showed that patients with IBD or psoriasis gained weight while receiving TNFi [25,26]. The weight gain ranged between 1.5 and 3 kg, and rarely exceeded 5 kg [25]. Weight gain was observed during the first six months of treatment (but as early as 4 weeks) and described with the initially approved TNFi, infliximab and etanercept, then with adalimumab. In parallel, BMI also increased. This phenomenon was then described in different immune-mediated diseases, RA, ax-SpA and PsA (Table 1). 

Numerous studies have examined the impact of TNFi on body composition in IJD (Table 1) [27,28,29,30,31,32,33,34,35,36,37,38]. The question of weight/BMI/body composition changes during TNFi was examined in patients with RA in 7 studies [27,28,29,30,31,32,33], in patients with SpA in 3 studies [35,36,37] and in patients with PsA in one study [38]. In one study, patients had RA or SpA [34]. Only two studies were randomized trials [27,30] while the others were open-labeled [28,29,31,32,33,34,35,36,37,38]. Collectively, a weight gain/BMI increase was reported by 6 studies [31,32,33,35,36,38], while fat mass increase was described in 7 [28,30,31,34,36,37,38]. No changes in fat mass were reported in 3 studies [27,34,36] with or without a parallel increase in lean mass [27,35,36]. In a randomized study comparing the effects of methotrexate (MTX), a csDMARD, and etanercept on body composition, there was no overall change in weight, BMI or body composition. However, in the etanercept group, 6/13 patients gained weight that was explained by an increase in lean mass [27]. Conversely, the study by Engvall et al. described increased body fat mass in patients with early RA while receiving infliximab while no similar changes were observed in the control group under csDMARDs [30]. Finally, a prospective observational study followed patients with RA starting a TNFi or MTX. After 24 months of therapy, weight gain was observed only among patients treated by TNFi [33].

#### 3.1.2. IL-6 Inhibitors

There are a limited number of studies examining changes in weight, BMI and body composition with IL-6i (Table 2) [39,40,41,42]. Tocilizumab, an anti-IL-6 receptor, was associated with weight gain during randomized controlled trials, and this has been confirmed in observational studies [39]. Three studies examined changes in body composition during TCZ treatment in patients with RA. One study with a short (3 month) follow-up found no variation in body composition parameters [40]. On the contrary, the two other studies reported an increase in weight and BMI that was explained by a significant increase in lean mass. Conversely, no change in fat mass was observed in these studies [41,42].

#### 3.1.3. Abatacept and Rituximab

There is no specific study examining this question, but neither abatacept nor ritixumab was associated with bodyweight changes during clinical trials. Bodyweight, BMI, and waist circumference did not change under abatacept IV given for 6 months in 15 patients with RA [43]. A study evaluated the changes in BMI in patients with antineutrophil cytoplasmic antibody-associated vasculitis while receiving RTX. Patients received 4 weekly infusions of RTX 375 mg/m^2^ followed by a placebo for a period of 18 months. The results showed that BMI significantly increased at 6 months (+1.1 ± 2.2 kg/m^2^), then remained stable without returning to baseline values. However, this increase in BMI was multifactorial and resulted from the improvement of disease activity, but also from glucocorticoid exposure and RTX administration. It was concluded that RTX also had effects on BMI independently of its impact on disease activity [44]. No similar data are available in patients with RA. 

#### 3.1.4. Anti-IL-23 and Anti-IL-17A Agents 

Ustekinumab, an anti-IL-23 p19i, has been associated with weight changes in patients with psoriasis. Indeed, this treatment increased BMI by 3.5% [45]. However, results are conflicting. In a prospective study comparing the effects of infliximab, a TNFi, and ustekinumab on body weight and BMI, only infliximab was associated with changes in these parameters [46]. DEXA assessment during ustekinumab treatment was not available, but a study evaluated body composition using bioelectrical impedance: 53 patients with psoriasis were evaluated at baseline and at 6 and 12 months. Decreases in body weight, BMI and fat mass were observed at month 6, while there was an increase at month 12 in body cell mass and phase angle, two indicators of nutritional status [47]. Secukinumab was not associated with changes in body weight or BMI in patients with psoriasis followed for 7 months after initiating the treatment [48]. However, a substantial weight gain (+10 kg) was reported in a patient with type 2 diabetes and PsA after 3 months of secukinumab treatment [49].

### 3.2. tsDMARDs

The effects of tofacitinib on anthropometric measurements and body composition have rarely been examined. In female mice, tofacitinib had no influence on body weight or body composition [50]. In a small series of female patients with RA (*N* = 4), it was also observed that tofacitinib (5 mg twice a day) did not affect body weight, while fat mass (evaluated by bioelectric impedance) slightly increased after 3 months of treatment [50]. In a larger series of patients with RA (*N* = 31) followed for 12 months after starting tofacitinib, a weight increase was observed (+4.2%), while BMI did not change significantly. Interestingly, this study showed that the visceral adiposity index, a mathematical model of visceral adipose tissue estimation, decreased during tofacitinib treatment [51]. There is no available study examining the changes in body composition under tofacitinib or baricitinib using DEXA. 

## 4. Influence of Weight or Body Mass Index on the Therapeutic Response to bDMARDs or tsDMARDs Used to Treat Inflammatory Joint Diseases

The association with obesity concerns different IJD [6], but this comorbidity mainly affects patients with PsA [52]. The impact of obesity, and body weight in general, on the therapeutic response to conventional, but also targeted drugs is a relevant issue [53]. The first data on the potential effects of obesity on the therapeutic response to bDMARDs came from psoriatic patients and showed that obese patients had a poor skin response to TNFi [54]. This question was further examined in different studies involving patients with RA and SpA, including PsA (Table 3 and Table 4). The therapeutic response to bDMARDs according to BMI was mainly examined with TNFi, and less with other bDMARDs. Limited data have been published with tsDMARDs. Results stem from retrospective studies, observational cohort studies or alternatively, from a limited number of randomized controlled trials.

### 4.1. bDMARDs in Rheumatoid Arthritis

#### 4.1.1. TNFi

Results are available for infliximab, etanercept, and to a lesser extent, for adalimumab and the other TNFi (Table 3). Patients had established disease except in one study [55]. 

The duration of follow-up varied from 4 to 12 months. Reported clinical outcomes were remission according to different definitions (disease activity score-28 joints (DAS-28), simplified disease activity index [SDAI] or clinical disease activity index (CDAI)), variation in DAS28 score, or time to discontinuation of the drug. All together, published studies show that in RA, obesity was present in 10% to 40%, and overweight in 32% to 57% [55,56,57,58,59,60,61]. There was clearly a worse clinical response to the different TNFi among obese or overweight patients compared to patients in the normal BMI group. This effect was independent of the route of administration of the biological drug (intravenously [IV] or subcutaneously [SC]) and observed for both fixed (etanercept and adalimumab) and weight-based doses (infliximab). The impaired response to TNFi in overweight/obese subjects was described regardless of the clinical outcome (remission or variation in DAS28). In one study, the lowest percentage of remission was observed with infliximab compared to etanercept or adalimumab [58]. One study examined the drug retention of a second line bDMARD following the failure of the first TNFi in obese patients. This study showed that the survival of the first TNFi was lower in obese (39.4%) compared to non-obese patients (49.1%). However, for patients who started a second non-TNFi biological drug, the persistence on therapy at a median period of 57 months was statistically lower in obese (43.5%) than in non-obese patients (80%). In that study, obese status was identified as a risk factor for discontinuing TNFi as first-line (Hazard Ratio (HR) and 95% CI: 1.64 (1.02–2.62)) and as second-line biological drug (HR: 2.9 1.08–8.45)) [60]. Only one study did not report a loss of efficacy of TNFi in obese patients. This was an analysis from the Veterans Affairs administrative database of US veterans with RA. In this large retrospective cohort including 23,669 subjects with RA, severe obesity (defined as a BMI > 35 kg/m^2^) was not associated with treatment discontinuation of the first SC administered TNFi, after adjustment for important covariables. On the contrary, patients with low (<20 kg/m^2^) or normal (20–25 kg/m^2^) BMI were more likely to discontinue a TNFi (and hydroxychoroquine and MTX as well) compared to overweight patients [61].

Data are more limited for the other bDMARDs.

#### 4.1.2. IL-6i

Two retrospective studies did not show any influence of body weight/BMI on the therapeutic response to tocilizumab (Table 3) [62,63].

#### 4.1.3. B- Cell Depletion Agent

For rituximab, a retrospective study also found no influence of BMI on the clinical outcome as evaluated by the decrease in DAS28 or percentage of good responders and remission [64]. However, in the study by Iannone et al., obese patients receiving rituximab as second-line bDMARD responded less well to this agent compared to normal-weight patients (Table 3) [60].

#### 4.1.4. Abatacept

Data stem from two retrospective studies and large prospective phase IV trials or registries (Table 3) [65,66,67,68,69]. Taken together, these studies showed that body weight or BMI did not influence response to abatacept given by the SC or IV route.

### 4.2. bDMARDs in Spondyloarthritis and Psoriatic Arthritis

In ax-SpA, analysis of the therapeutic response according to body weight was limited to TNFi (Table 4). Four retrospective studies were published, and as for RA, the therapeutic response (as evaluated by Ankylosing Spondylitis Disease Activity Score (ASDAS) or rate of Assessment of Spondyloarthritis Society (ASAS) responders) was impaired in obese or overweight patients [70,71,72,73]. Results were mainly given for infliximab. The duration of the studies ranged from 6 to 12 months. 

Four studies examined therapeutic response in patients with PsA taking into account their body weight/BMI (Table 4) [74,75,76,77]. Clinical outcomes were the achievement of minimal disease activity (MDA), DAS28 (or SDAI) response and the European League Against Rheumatism (EULAR) response. The duration of follow-up was 6 to 36 months. As previously observed in RA and ax-SpA, BMI influenced the outcome, with fewer responders in the obese category. However, in one study, the disease remission rate based on DAS28 or SDAI did not differ between obese, overweight and normal-weight patients [75].

For ustekinumab, the IL-23p19i, data from psoriatic patients showed that obesity impaired the skin response as evaluated by the psoriasis area severity index (PASI) [78,79]. Thus, the recommended dose for patients weighing 100 kg or more has been fixed at 90 mg per injection [80]. There are no data on the efficacy of ustekinumab and secukinumab in patients with PsA according to body weight. In a pooled analysis of phase III trials in psoriasis evaluating secukinumab, response rates were analyzed by body-weight quartiles. The results showed that responses remained meaningful in all weight groups, although there was a trend towards a lower response by increasing body weight [81,82]. However, secukinumab 300 mg dose demonstrated a greater benefit than 150 mg dose.

Abatacept is also licensed for the treatment of PsA. In a post hoc analysis of phase III trials, it was concluded that BMI did not affect clinical or radiographic response to SC abatacept in patients wih PsA [83].

4.3/data on bDMARDs response from meta-analysis and registry: Two meta-analyses examined the association between obesity and response to TNFi or different bDMARDs in IJD [84,85]. The first was based on 54 cohorts and included patients with various immune-mediated diseases, such as IJD but also psoriasis and IBD. Only the response to TNFi was analyzed. This analysis showed that patients in the obese category had a 60% higher risk of failing therapy (Odds Ratio (OR) (95% CI):1.6 (1.39–1.83)). A dose-response relationship was observed, i.e., the higher the body weight, the lower the response. These effects were observed in the different immune-mediated diseases, except for IBD [84]. The second meta-analysis examined the efficacy of all bDMARDs used in IJD and was based on 24 articles, including 10 on RA, 4 on ax-SpA, 2 on PsA and 8 on psoriasis and Crohn’s disease. This meta-analysis showed that the odds of achieving a good response or remission were lower in obese than in non-obese patients who were treated by TNFi (good response in RA, OR: 0.34 (0.18–0.64); remission in RA: OR: 0.36 (0.21–0.59); Bath Anlylosing Spondylitis Disease Activity Index 50 (BASDAI50) in ax-SpA: OR 0.41 (0.21–083)). No significant difference between obese and non-obese patients was found for those treated by abatacept or tocilizumab [85].

Lastly, data from the German biologic registry RABBITT were recently published. This registry included data from 10,593 patients with RA and the influence of obesity was analyzed after adjustment for potential confounders. Obesity was found to have a negative effect on the improvement of DAS28 for women receiving a csDMARD or a TNFi, but also for women and men receiving tocilizumab. Conversely, obesity did not impact on treatment efficacy with rituximab and abatacept [86].

**Table 4 metabolites-10-00107-t004:** Response to bDMARDs in obese/overweight patients with spondyloarthritis.

Author (Reference)	Number of Subjects	Disease	Sex Ratio M/F (%)	Disease Duration (Years)	Obese/Overweight Subjects (%)	bDMARDs	Outcome	Main Results
Ottaviani [70]	155	AS	63.3/36.7	8	overweight: 35 obese: 25	IFX (5 mg/kg)	BASDAI50 at month 6	Fewer responders in obese/overweight groups
Gremese [71]	170	ax-SpA	69.4/ 30.6	16.3	overweight: 32.4 obese: 13.5	IFX (5 mg/kg) or ETA or ADA	BASDAI 50 month 6	Rate of responders lower in obese and overweight patients
Micheroli [72]	624	ax-SpA	62.2/ 37.8	13	overweight: 32.7 obese: 14.1	all TNFi	Rate of ASAS40 responders at one year	Rate of responders lower in obese and overweight patients
Ibanez Vodnizza [73]	41	AS	61/39	14.6	overweight: 36.6 obese: 12.2	ETA or ADA	BASDAI or ASDAS-CRP change at month 6	Higher body fat associated with worse response to TNFi
Di Minno [74]	270	PsA	62/38	9.2	obese: 50	IFX (5 mg/kg) ETA or ADA	MDA at month 12	Rate of MDA < obese patients vs non obese
Iannone [75]	135	PsA	50.4/49.6	ND	overweight: 34.8 obese: 33	IFX (5 mg/kg) ETA or ADA	DAS28 or SDAI response	No difference in rate of remission according to DAS28 or SDAI
Eder [76]	557	PsA	58.4/41.6	15	overweight: 36.2 obese: 35.4	TNFi without precision	MDA at month 12	Less MDA in obese category
Hojgaard [77]	1943	PsA	44.5/55.5	4	obese: 34.6	all TNFi	EULAR response at month 6	EULAR response lower in the obese category
Mc Innes [83]	422	PsA	female % according to BMI categories: - placebo: 41% to 64% abatacept: 42.9 to 67.7%	NA	Placebo group: overweight: 27.1% obese: 54.3% Abatacept group: overweight: 36.3% obese: 49%	abatacept SC	ACR20	no difference in the rate of responders between obese/overweight and normal weight patients

(TNF inhibitor, IL-23 inhibitor and IL-17A inhibitor) in axial spondyloarthritis and psoriatic arthritis according to body weight or body mass index (M: male; F: female; BMI: body mass index; bDMARD: biological disease-modifying antirheumatic drug; TNFi: TNF inhibitor; IFX: infliximab; ETA: etanercept; ADA: adalimumab; AS: ankylosing spondylitis; SpA: spondyloarthritis; ax-SpA: axial spondyloarthritis; PsA: psoriatic arthritis; BASDAI: Bath Ankylosing Spondylitis Disease Activity Index; ASAS: Assessment of SpondyloArthritis Society; EULAR: European League Against Rheumatism; MDA: minimal disease activity; ACR: American College of Rhematology; NA: not available).

### 4.3. tsDMARDs 

In a post hoc analysis of baricitinib trials in RA, certain baseline patient characteristics were identified to have an effect on the drug response. They included specific ethnic group (non-Asian and non-white), negativity for rheumatoid factors and anti-citrullinated protein antibodies but also weight more than 100 Kg [87].

## 5. Discussion

The treatments that are routinely given in IJD may have specific consequences on body weight, BMI or body composition. This impact is well described with certain bDMARDs, and less so with tsDMARDs. The most demonstrative data are well illustrated by weight gain, and BMI and fat mass changes observed under TNFi. This effect is observed with the different TNFi (albeit mainly with infliximab) and in the different forms of IJD [88]. Weight gain is predominantly explained by a gain of fat that localized toward the android/abdominal (visceral) region [35,36]. In parallel, there was also a gain of lean mass, which is less sizable [28]. TNFα is a catabolic cytokine that plays a crucial role in the development of cachexia, a state that is well described in RA [10]. Indeed, TNFα, in concert with other pro-inflammatory cytokines, induces proteolysis and lipolysis. The administration of TNFi thus controls the negative effects of TNFα on cell mass and muscle mass. In this sense, the randomized trial by Marcora et al. clearly demonstrated a specific effect of TNFα inhibition on fat-free mass [27]. The increase in lean mass is modest and can be explained by the control of inflammation and disease activity, resulting in improved well-being and physical activity. Modulation of appetite and gut/growth hormones may also play a role [89]. The predominant and significant gain of fat mass under TNFi may also be explained by the control of inflammation and disease activity. The fat mass increase was small, but significant and located specifically towards the abdominal/visceral region, a type of adipose tissue strongly associated with enhanced CV risk [7]. This effect is drug-specific, since it was not observed with a csDMARD (MTX) in the study by Engvall et al. [30]. Indeed, visceral fat mass is associated with metabolic complications, atherosclerosis development and adverse CV events [7]. These changes in body composition during TNFα inhibition with the increase in abdominal/visceral fat raised questions about the ultimate effect on CV risk in patients receiving this class drug. However, reassuring data have now emerged regarding the long term protective effects of TNFi on CV diseases [88,89,90]. Indeed, it is thought that the effective control of disease activity and thus of chronic inflammation in IJD have a favorable impact on the CV prognostic of the patients. This is well suggested by the reduced risk of CV events for patients with RA under TNFi [91]. 

The IL-6i tocilizumab has been associated with a significant gain in lean mass without modification of fat, counteracting the process of cachexia in patients with RA [41,42]. This effect may be explained by the specific properties of IL-6 on muscle physiology. Thus, IL-6 inhibition may have anabolic effects on myocytes [92]. IL-6, in conjunction with TNFα may promote muscle loss and is associated with a decline in muscle mass and muscle strength, especially in older people. Indeed, elevated serum IL-6 has been associated with frailty and physical function in the aging population [93]. In addition, muscle atrophy has been observed in IL-6 transgenic mice and this was reversed by IL-6 receptor blockade [94]. 

Co-stimulatory inhibition with abatacept does not induce changes in weight/BMI, while for the B-cell depletion agent rituximab, the question remains open. Indeed, in vasculitis, this latter agent was associated with an increase in BMI that may be multifactorial [44]. For the IL-23/th17 pathway and related cytokine inhibitors, both ustekinumab and anti-IL-17Ai seem to have neutral effects on weight and BMI, but more detailed evaluations of body composition with DEXA are warranted. 

The JAKi tofacitinib induces weight gain and seems to have a beneficial effect on fat mass, by reducing visceral adipose tissue [51]. However, specific studies examining body composition and fat mass under the different JAKi, tofacitinib and baricitinib, are required. 

In parallel, obesity has a clear impact on disease outcomes in IJD, and on the therapeutic response to various treatments. bDMARDs are used at a weight adapted-dose only for those given by the IV route (infliximab, tocilizumab or abatacept), while for subcutaneously administered biologics, a fixed-dose is recommended. However, for ustekinumab and golimumab, there are recommendations to increase the drug dosage (respectively from 45 to 90 mg and from 50 to 100 mg) for patients weighing more than 100 kg. There is robust clinical evidence from observational studies and the real-life setting that biological agents are less effective in obese and even overweight patients and this is especially well demonstrated for TNFi [84,85,86]. This negative effect was reported in the different forms of IJD, mainly with infliximab, and was then reported with etanercept and adalimumab. Results on the specific response to certolizumab and golimumab according to body weight are less well documented. Based on the above literature review and the 2 meta-analyses, some key observations related to the influence of body weight/obesity can be made. The clinical response to TNFi in IJD was globally impaired in obese subjects, regardless of the outcome measurement. There was a dose-dependent effect in the sense that each unit increase in BMI is associated with an increase in the odds of failing therapy (by a factor of 6.5 according to the meta-analysis by Singh et al.) [84]. This resistance to TNFi is independent of the route of administration of the treatment [84]. The results for other bDMARDs are more limited, but indicate that response to tocilizumab is probably not influenced by body weight [62,63]. However, the SUMMACTA trial showed that tocilizumab given SC is less effective in obese compared to non-obese patients [95]. Furthermore, in the German RABBIT registry, response to tocilizumab was also attenuated by obesity [86]. Additional studies are thus required for this biological agent. For abatacept, results are robust for both the IV and SC routes, and show no influence of bodyweight/BMI on the efficacy of the drug [65,66,67,68,69]. For the B cell depletion agent rituximab, results are contradictory, and showed no effect of weight in one retrospective analysis [64], while the opposite was reported in a second study [60]. For ustekinumab and IL-17A agents, data examining the influence of obesity on the therapeutic efficacy are scare and even lacking in IJD [96]. Data come from psoriatic studies and reported a clear effect of obesity on ustekinumab response. This explains why the dose of this drug must be adapted for patients weighing more than 100 kg [80]. The clinical response to the anti-IL-17A agent secukinumab in IJD does not seem to be influenced by body weight but more data are needed. According to these results, for patients with high BMI or obese status, the clinician may be orientated for selecting a weight-adapted targeted DMARD. Actually, TNFi does not appear to be the best choice. In RA, abatacept or rituximab may be proposed or alternatively an IL6i. In ax-SpA, IL-17Ai may be a preferred choice. In PsA, the prescribed dose may be adapted for ustekinumab and golimumab, but IL-17Ai may be another option. 

All these data show that obesity may be a factor for resistance to TNFi, an effect that is less pronounced for tocilizumab and rituximab, and not observed for abatacept. The reasons why the TNFi biological class selectively fails to respond in obese patients are unknown. Multiple explanations have been proposed such as the pharmacokinetic properties and the volume of distribution of the drugs, taking account of their lipophilic properties [6,53]. Obesity is a state that modifies the pharmacokinetics of TNFi by increasing the clearance of the drug, resulting in a shorter half-life and lower serum trough drug concentrations [84]. In parallel, obesity and overfat are conditions that may perpetuate chronic systemic low-grade inflammation, through increased levels of cytokines, chemokines and adipokines [6]. Adipokines such as leptin and adiponectin are involved in immune response and they may contribute to the inflammatory process in various immune-mediated diseases [97]. Therapeutic resistance to TNFi is independent of the route of administration and observed even for weighted-dose biologics such as infliximab. Infliximab also seems to work less well compared to fixed-dose administered drugs such as etanercept or adalimumab [58]. One potential explanation is that omental adipocytes have high expression of Fc receptors and the presence of this receptor modulates the efficacy of infliximab in both in vitro and ex vivo models [6]. A specific analysis of the therapeutic response to certolizumab (a TNFi without an Fc fragment) according to body weight and fat mass is thus a relevant issue. One weakness of the different studies into the association between obesity and therapeutic response is that the analyses were performed according to body weight (or BMI) and not fat mass. The study by Ibanez Vodnizza et al. specifically analyzed the TNFi clinical response in patients with ax-SpA according to fat mass [73]. This study confirmed that fat mass was associated with impaired clinical response: higher body weight, BMI, body fat percentage, fat mass and fat mass index at baseline were all associated with a lower chance of having a clinical improvement according to ASDAS-CRP change. The multivariate analysis confirmed that fat mass was a significant negative predictor of achieving a clinical response to TNFi in ax-SpA. 

Finally, we have some information on the impact of diet interventions in obese patients with IJD or other immune-mediated diseases. Such interventions are beneficial, showing that weight loss improves treatment response to TNFi. In obese/overweight patients with PsA starting a TNFi, a hypocaloric diet was associated with more patients in MDA than patients with a free managed diet. Weight loss of more than 5% was predictive of the achievement of MDA [98]. In addition, weight loss in patients with PsA or psoriasis under stable treatment improved the different measurements of (skin and joint) disease activity [99,100]. Bariatric surgery also improves the disease activity of patients with RA [101].

We focus our review on the most common IJD. However, the crosstalk between adipose tissue and response to therapy also involved other immune-mediated diseases such as Crohn’s disease or psoriasis [54,84]. Obesity worsens the course of systemic lupus erythematosus. However, limited information is available on the relationship between the therapeutic response to conventional treatments and BMI [102]. 

## 6. Conclusions

The relationships between obesity/adipose tissue and bDMARDs/tsDMARDs in IJD are bidirectional. On the one hand, certain bDMARDs induce changes in body composition, especially TNFi, with an increase in fat mass and minor changes in lean mass. Tocilizumab is also involved by selectively increasing lean mass. On the other hand, therapeutic response to bDMARDs is hampered in obese patients, especially with TNFi. Obesity is thus a relevant comorbidity to consider in patients with IJD. Obesity may be accentuated by TNF inhibition, at least to a moderate degree, but this drug class favors visceral adiposity. On the contrary, IL-6 inhibition has a more beneficial effect by increasing lean mass and skeletal muscle, counteracting cachexia. More data on the effects of the other bDMARDs and JAKi on body composition are needed. The finding that obesity compromises the therapeutic efficacy of certain bDMARDs (TNFi and, with less evidence, rituximab or tocilizumab) but not others (abatacept), may help the physician with personalized medicine and guide the therapeutic choice in obese/overweight patients. Weight management also appears to be crucial in obese patients to improve disease activity. Specific studies using DEXA examining the direct impact of obesity, and, ideally, of fat mass, on the therapeutic response of recently licensed bDMARDs and tsDMARDs are required. 

## Figures and Tables

**Table 1 metabolites-10-00107-t001:** Changes in body composition during Tumour necrosis factor (TNF) inhibitor treatment in inflammatory rheumatic diseases.

Author (Reference)	Patients (N)	Age, Years	Disease Duration	Inflammatory Joint Disease	Sex Ratio	Study Design	Study Duration	Tnfi	Body Composition Assessment	Results
Marcora [27]	26	52	<6 months	RA	18 F, 6 M	randomized phase 2 trial	6 months	ETA or MTX	DEXA	No overall changes. Weight gain in 6/13 patients in ETA group: gain in lean mass
Metsios [28]	20	61.1	17.3 years	RA	10 F, 10 M	open label	12 weeks	ND	bioelectrical impedance	↑ truncal fat mass
Serelis [29]	19	54	5 months	RA	19 F	open label	1 year	IFX or ADA	DEXA	No changes in fat or lean mass
Engvall [30]	40	57.5	5 months	RA	29 F, 11 M	randomized trial	2 years	IFX or HCQ and SLZ	DEXA	↑ fat mass [+ 3.4 kg) and fat mass index in the IFX group
Kopec- Medrek [31]	16	ND	7.1 years	RA	16 F	open label	12 months	IFX	DEXA	↑ weight, BMI and fat mass
Chen [32]	20	53.8	6.6 years	RA	18 F, 2 M	control group without ETA	12 months	ETA	bioelectric impedance	↑ weight and BMI in the ETA group. No changes in fat mass
Toussirot [34]	20	48.6	9.6 years	8 RA, 12 ax-SpA	14 F, 6 M	open label	2 years	IFX, ETA or ADA	DEXA	↑ fat mass, ↑ % fat, ↑ fat android region, and visceral fat
Briot [35]	19	21–71 (median: 40)	16.5 years	SpA (10 ax-SpA, 4 PsA, 5 IBD)	2F, 17 M	open- label	12 months	IFX 3–5 mg/kg	DEXA	↑ weight + 2.2 kg, ↑ lean mass + 1.4 kg; no change in fat mass
Briot [36]	106	38	1.5 years	SpA (60 ax-SpA, 46 p-SpA)	26 F, 80 M	open label	2 years	IFX or ETA	DEXA	↑ body weight (+3.5%), ↑ fat mass (+14.5%); ↑ lean mass + 2%)
Hmamouchi [37]	65	39.3	13.1	ax-SpA	22 F, 63 M	open label	2 years	IFX or ETA	DEXA	↑ subcutaneous and visceral fat
Renzo [38]	20	42.2	14.1 years	PsA	ND	open label	6 months	IFX or ETA	DEXA	↑ weight (+2.1%) and BMI. ↑ fat mass (+8.9% and lean mass (+2.9%)

(M: male; F: female; BMI: body mass index; TNFi: TNF inhibitor; IFX: infliximab; ETA: etanercept; ADA: adalimumab; MTX: methotrexate; HCQ: hydroxychloroquine; SLZ: sulfasalazine; DEXA: dual X-ray absorptiometry; RA: rheumatoid arthritis; SpA: spondyloarthritis; ax-SpA: axial spondyloarthritis; pSpA: peripheral spondyloarthritis; PsA: psoriatic arthritis; IBD: inflammatory bowel disease).

**Table 2 metabolites-10-00107-t002:** Changes in body composition during tocilizumab treatment in rheumatoid arthritis.

Author (Reference)	Patients (N)	Age (Years)	Disease Duration	Inflammatory Joint Disease	Sex Ratio	Study Design	Study Duration	IL-6i	Body Composition Assessment	Results
Younis [39]	21	52	NA	RA	NA	controlled trial	4 months	21 TCZ, 16 IFX	NA	↑ body weight and BMI (+0.3 unit). No changes under IFX
Hugo [40]	16	57.2	13 years	RA	14 F, 2 M	open label	3 months	TCZ	DEXA	No changes in body weight, body composition
Tournadre [41]	21	57.8	8.5 years	RA	16F, 4 M	open label	12 months	TCZ	DEXA	↑ body weight, ↑ BMI, ↑ lean mass and fat free mass. No change in fat mass
Toussirot [42]	106	56.6	9.9 years	RA	78 F, 28 M	open label	12 months	TCZ	DEXA	↑ BMI, ↑ lean mass. No change in fat mass

(RA: rheumatoid arthritis; BMI: body mass index; M: male; F: female; IL6i: interleukin-6 inhibitor; TCZ: tocilizumab; IFX: infliximab; DEXA: dual X-ray absorptiometry; NA: not available).

**Table 3 metabolites-10-00107-t003:** Response to biological agents (bDMARDs) in obese/overweight patients with rheumatoid arthritis.

Author (Reference)	Number of Subjects	Sex Ratio M/F (%)	Disease Duration (Years)	Obese/Overweight Subjects (%)	bDMARDs	Outcome	Main Results
Heimans [55]	508	32/68	0.4	obese and overweight: 57.4	csDMARDs combination or MTX + IFX	DAS≤ 2.4 at one year	RR DAS≤ 2.4 at one year in obese /overweight patients: 2.2 [0.99–4.92]
Klaasen [56]	89	23/77	7	obese: 16.8	IFX [3 mg/kg)	∆DAS28 week 16	Negative relationship between BMI and ∆DAS28 or remission
Smolen [57]	761	17/83	6.9	overweight: 35.2 obese: 16.7	ETA (50 mg/week)	remission (according to DAS28, CDAI, SDAI) week 36	Negative relationship between BMI and remission
Gremese [58]	641	19/81	8.4	overweight: 32.3 obese: 10.3	IFX(3mg/kg), ETA (50 mg/week) or ADA (40 mg eow)	remission (DAS28) at one year	% remission in obese/overweight patients < normal BMI
Ottaviani [59]	76	17/83	8	overweight: 38.2 obese: 28.9	IFX (3 mg/kg)	∆DAS28 ≥ 1.2 at 6 months	Fewer responders in overweight/obese patients compared to normal BMI
Iannone [60]	292	15/85	12	overweight: 37.3 obese: 22.6	All bDMARDs at the second line of treatment	drug survival at one year	Less drug persistence in obese patients versus normal weight
McCulley [61]	23,669	87/13	NA	overweight: 36.6 obese: 40.1	All csDMARDs and subcutaneous injectable TNFi	time to treatment discontinuation	Severe obesity not associated with treatment discontinuation compared to overweight BMI for all except prednisone. Low BMI was associated with TNFi discontinuation
Pers [62]	222	17.6/82.4	14	overweight: 26 obese: 15.5	TCZ IV (8 mg/kg)	EULAR response at month 6	Similar response according to BMI
Gardette [63]	115	84.3/15.7	11	overweight: 32 obese: 22	TCZ IV (8 mg/kg)	∆DAS28 ≥ 1.2 at month 6	No effect of BMI on TCZ response
Ottaviani [64]	114	81.5/18.5	9.6	overweight: 35.9 obese: 30	RTX IV (1 g × 2 days 1 and 15)	∆DAS28 ≥ 1.2 at month 6	No effect of BMI on RTX response
Gardette [65]	141	82.3/17.7	12.5	overweight: 27 obese: 27.6	ABA IV (500 mg < 60 kg; 750 mg 60–100 kg; 1000 mg > 100 kg)	∆DAS28 ≥ 1.2 at month 6	No effect of BMI on ABA response
D’Agostino (ACQUIRE trial) [66]	1456	82.5/17.5	7.6	overweight: 34 obese: 30	ABA IV or SC	Remission (DAS28 < 2.6) at month 6	Rate of remission similar across BMI groups
Mariette (ACTION study) [67]	643	73.8/26.2	7.2	overweight: 35 obese: 24	ABA IV (500 mg < 60 kg; 750 mg 60–100 kg; 1000 mg > 100 kg)	Drug retention at 6 months	Retention rates similar across BMI groups
Di Carlo [68]	130	83.8/16.2	11.2	NA	ABA SC or IV	DAS28 remission or Boolean criteria for remission	No effect of BMI on ABA response
Ianone (PANABA registry) [69]	2015	80.6/19.4	10.2	obese: 18.9	ABA IV (500 mg < 60 kg 750 mg 60–100 kg; 1000 mg > 100 kg)	Drug retention of ABA	No difference in ABA retention between obese and non obese patients

(TNF inhibitor, IL-6 inhibitor, co-stimulatory inhibitor and B-cell depletion agent) in rheumatoid arthritis according to body weight or body mass index (M: male; F: female; BMI: body mass index; bDMARD: biological disease-modifying antirheumatic drug; csDMARD: conventional synthetic disease modifying antirheumatic drug; IFX: infliximab; ETA: etanercept; ADA: adalimumab; TCZ: tocilizumab; ABA: abatacept; RTX: rituximab; MTX: methotrexate; RA: rheumatoid arthritis; DAS28: disease activity score 28 joints; SDAI: simplified disease activity score; CDAI: clinical disease activity index; RR: relative risk; eow: every other week; IV: intravenous; SC: subcutaneous; NA: not available).

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
