# Peer review of "The Interrelations between Biological and Targeted Synthetic Agents Used in Inflammatory Joint Diseases, and Obesity or Body Composition"

_metabolites, 2020, doi:10.3390/metabo10030107_

Round 1

Reviewer 1 Report

This review by Eric Toussirot describes the relation between obesity and the treatment for rheumatic joint diseases. The review is clear and well written and the topic is thoroughly addressed.

A few points to improve the review:

- paragraph 3.1.1 TNF inhibitors: This paragraph is difficult to follow. I suggest that the author tries to summarize the major points in common between the different studies (as done in paragraph 4.1.1) instead than describing the single studies

- discussion, page 21 line 365: I would appreciate a couple more sentences on the relation between TNFi and cardiovascular risk.

Minor points:

- after “TNF” there is always space before the following word

- page 4 line 154: the text does not refer to Table 2

- page 11, line 232: Please rephrase the following sentence to make it clearer: “The association with obesity concerns different IJD (6), but seems most relevant in PsA”

- page 20, line 319: I would suggest to separate this paragraph (where the author discusses about two big meta-analyses) adding a different title here.

Author Response

Reviewer 1 :

This review by Eric Toussirot describes the relation between obesity and the treatment for rheumatic joint diseases. The review is clear and well written and the topic is thoroughly addressed.

A few points to improve the review:

- paragraph 3.1.1 TNF inhibitors: This paragraph is difficult to follow. I suggest that the author tries to summarize the major points in common between the different studies (as done in paragraph 4.1.1) instead than describing the single studies

Response : as suggested, the major results were summarized and the studies were ordered in Table 1 by disease (RA, SpA and PsA)

discussion, page 21 line 365: I would appreciate a couple more sentences on the relation between TNFi and cardiovascular risk.

Response : as requested CV risk in IJD was commented and a reference was added (effect of TNFI on CV risk in IJD, méta-analysis by Roubille et al)

Minor points:

- after “TNF” there is always space before the following word

Response : yes, we agree

- page 4 line 154: the text does not refer to Table 2

Response : it was removed

- page 11, line 232: Please rephrase the following sentence to make it clearer: “The association with obesity concerns different IJD (6), but seems most relevant in PsA”

Response : the phrase was changed

page 20, line 319: I would suggest to separate this paragraph (where the author discusses about two big meta-analyses) adding a different title here.

Response : a new section was added

Reviewer 2 Report

Obesity can be a risk factor for various inflammatory joint diseases (IJD). Treatment using DMARDs (biologics or targeted synthetics) may further alter body weight, as well as body composition. In turn, obesity has been evidenced to influence therapeutic efficacy or failure. Hence, a good understanding of the relationship between body composition/obesity and IJD therapy is crucial to inform a personalized, effective therapeutic strategy. This review was thoughtfully put together to serve as a physician’s guide to understand this aspect of disease management and help optimize personalized medicine. The author has extensive experience in the field and put together a broad, thorough overview.

While the review succeeds in answering the main issues, a few questions remain. While the tables effectively summarize any relevant research, the order in which papers are cited seems quite random. The tables might be easier to interpret if the order is more purposeful, for example different types of IJD grouped together (e.g. table 4) or studies with similar findings, or using the same therapeutics could follow each other directly (e.g. table 1). I also wondered why the review discusses in detail all implications for so-called bDMARDs and tsDMARDs, but not csDMARDs.

Some specific comments:

(1)        On line 135, please clarify when you switch and start talking about tsDMARDs instead of bDMARDS. It would also be helpful to point out in this paragraph how tsDMARDs differ from csDMARDs.

(2)        Lines 164-168: First sentence states etanercept does not have an effect on overall weight, BMI or body composition, while the next sentence says the weight gain is explained by an increase in lean mass. These two sentences seem to contradict each other.

(3)        Why are the findings on line 312 (secukinumab) and line 317 (abatacept) not incorporated in table 4?  

(4)        The review focused on RA, ax-SpA/AS and PsA as the most common IJD. Could you briefly comment on whether similar findings have been reported in other IJD or whether this crosstalk between obesity and therapy has only been studied in the discussed pathologies?

(5)        Since the general aim is to advise therapeutic strategy, could the authors, based on the referenced research, put together a type of decision tree that instructs the best choice of therapeutic based on type of IJD, BMI, sex, etc.?

Author Response

Reviewer 2 :

Obesity can be a risk factor for various inflammatory joint diseases (IJD). Treatment using DMARDs (biologics or targeted synthetics) may further alter body weight, as well as body composition. In turn, obesity has been evidenced to influence therapeutic efficacy or failure. Hence, a good understanding of the relationship between body composition/obesity and IJD therapy is crucial to inform a personalized, effective therapeutic strategy. This review was thoughtfully put together to serve as a physician’s guide to understand this aspect of disease management and help optimize personalized medicine. The author has extensive experience in the field and put together a broad, thorough overview.

While the review succeeds in answering the main issues, a few questions remain. While the tables effectively summarize any relevant research, the order in which papers are cited seems quite random. The tables might be easier to interpret if the order is more purposeful, for example different types of IJD grouped together (e.g. table 4) or studies with similar findings, or using the same therapeutics could follow each other directly (e.g. table 1).

Response : as resquested, the studies were ordered in Table 1 by disease (RA, SpA and PsA)

I also wondered why the review discusses in detail all implications for so-called bDMARDs and tsDMARDs, but not csDMARDs.

Response : this review article covered the relationships between fat mass/obesity and the effects of bDMARDs/tsDMARDs on body composition and the therapeutic response to these drugs according to BMI. The therapeutic response to csDMARDs was not specifically discussed and this is mentionned in the introduction. In fact , therapeutic response to csDMARDS may be impaired by obesity (for instance MTX) but conversely, csDMARDs do not influence body composition. We are also limited by the length of the paper.    

Some specific comments:

(1)        On line 135, please clarify when you switch and start talking about tsDMARDs instead of bDMARDS. It would also be helpful to point out in this paragraph how tsDMARDs differ from csDMARDs.

Response : yes we agree and this is corrected

(2)        Lines 164-168: First sentence states etanercept does not have an effect on overall weight, BMI or body composition, while the next sentence says the weight gain is explained by an increase in lean mass. These two sentences seem to contradict each other.

Response : we clarify the results of this study

(3)        Why are the findings on line 312 (secukinumab) and line 317 (abatacept) not incorporated in table 4?  

Response : only results of the abatacept study in PsA was added in Table 4. For secukinumab, results are available in psoriatic patients but not PsA

(4)        The review focused on RA, ax-SpA/AS and PsA as the most common IJD. Could you briefly comment on whether similar findings have been reported in other IJD or whether this crosstalk between obesity and therapy has only been studied in the discussed pathologies?

Response : we add a brief commentary on similar findings in patients with psoriasis, Crohn’s disease and also systemic lupus.

(5)        Since the general aim is to advise therapeutic strategy, could the authors, based on the referenced research, put together a type of decision tree that instructs the best choice of therapeutic based on type of IJD, BMI, sex, etc.?

Response : in the discussion, we add a brief therapeutic decision tree in obese patients according to the disease